# Position representations of moving objects align with real-time position in the early visual response

**Philippa Anne Johnson[1]\*[†], Tessel Blom[1†], Simon van Gaal[2‡], Daniel Feuerriegel[1†], Stefan Bode[1†], Hinze Hogendoorn[1†]**

[1]University of Melbourne, Melbourne, Australia; [2]University of Amsterdam, Amsterdam, Netherlands

**Abstract** When interacting with the dynamic world, the brain receives outdated sensory information, due to the time required for neural transmission and processing. In motion perception, the brain may overcome these fundamental delays through predictively encoding the position of moving objects using information from their past trajectories. In the present study, we evaluated this proposition using multivariate analysis of high temporal resolution electroencephalographic data. We tracked neural position representations of moving objects at different stages of visual processing, relative to the real-time position of the object. During early stimulus-evoked activity, position representations of moving objects were activated substantially earlier than the equivalent activity evoked by unpredictable flashes, aligning the earliest representations of moving stimuli with their real-time positions. These findings indicate that the predictability of straight trajectories enables full compensation for the neural delays accumulated early in stimulus processing, but that delays still accumulate across later stages of cortical processing.

**\*For correspondence:**
pajohnson@student.unimelb.
edu.au

**Present address:** [†]Melbourne School of Psychological Sciences, University of Melbourne, Melbourne, Australia; [‡]ABC Amsterdam Brain and Cognition, University of Amsterdam, Amsterdam, Netherlands

## Editor's evaluation

This article presents compelling visual motion processing data generated via clever analyses, to shed fundamental insights into how the brain compensates for neural processing delays. The approach and findings will be of interest to all neuroscientists – where neural delays prove a sticking point for many theories. One of its major strengths lies in the successful attempt to generalize neural representations of static objects to moving objects.

## Introduction

Responding quickly to the world around us is a primary function of the central nervous system: catching prey, escaping predators, and avoiding moving objects (e.g. falling rocks) are all crucial to survival. This task is complicated by delays that accumulate during the neural transmission of information from the sensory organs to the brain. As a result, the brain only has access to outdated sensory information. Furthermore, additional delays accumulate during subsequent cortical processing of information. The world will inevitably change during this time, so how can the brain overcome this fundamental problem and keep up with an ever-changing world?

Several lines of evidence suggest that the brain can compensate for neural transmission delays through prediction: using information from the past to predict what is happening in the present (*Nijhawan, 1994*; *Kiebel et al., 2008*; *Friston et al., 2010*). Indeed, many researchers consider prediction to be a core objective of the central nervous system (*Friston, 2010*; *Clark, 2013*). This is relevant to all sensory processing, whether anticipating haptic input while manipulating an object

**eLife digest** The survival of animals depends on their ability to respond to different stimuli quickly and efficiently. From flies fluttering away when a swatter approaches, to deer running away at the sight of a lion to humans ducking to escape a looming punch, fast-paced reactions to harmful stimuli is what keep us (and other fauna) from getting injured or seriously maimed. This entire process is orchestrated by the nervous system, where cells called neurons carry signals from our senses to higher processing centres in the brain, allowing us to react appropriately.

However, this relay process from the sensory organs to the brain accumulates delays: it takes time for signals to be transmitted from cell to cell, and also for the brain to process these signals. This means that the information received by our brains is usually outdated, which could lead to delayed responses. Experiments done in cats and monkeys have shown that the brain can compensate for these delays by predicting how objects might move in the immediate future, essentially extrapolating the trajectories of objects moving in a predictable manner. This might explain why rabbits run in an impulsive zigzag manner when trying to escape a predator: if they change direction often enough, the predator may not be able to predict where they are going next.

Johnson et al. wanted to find out whether human brains can also compensate for delays in processing the movement of objects, and if so, at what point (early or late) in the processing pipeline the compensation occurs. To do this, they recorded the electrical activity of neurons using electroencephalography from volunteers who were presented with both static and moving stimuli. Electroencephalography or EEG records the average activity of neurons in a region of the brain over a period of time.

The data showed that the volunteers' brains responded to moving stimuli significantly faster than to static stimuli in the same position on the screen, essentially being able to track the real-time position of the moving stimulus. Johnson et al. further analysed and compared the EEG recordings for moving versus static stimuli to demonstrate that compensation for processing delays occurred early on in the processing journey. Indeed, the compensation likely happens before the signal reaches a part of the brain called the visual cortex, which processes stimuli from sight. Any delays accrued beyond this point were not accommodated for.

Johnson et al. clearly demonstrate that the human brain can work around its own shortcomings to allow us to perceive moving objects in real time. These findings start to explain, for example, how sportspersons are able to catch fast-moving balls and hit serves coming to them at speeds of approximately 200 kilometres per hour. The results also lay the foundation for studying processing delays in other senses, such as hearing and touch.

or auditory input while listening to a melody. While experimental paradigms in cognitive neuroscience often remove the dimension of time by using artificially static displays (*Millidge et al., 2022*), delays are a fundamental obstacle in neural processing and should be accounted for in any comprehensive theory of brain function. This problem has received particular interest within the field of motion perception, as the future positions of a predictably moving object can be determined by the object's current acceleration, velocity, and position. In this case, there is evidence that predictive processes help to compensate both for the neural delays incurred before visual input reaches the visual cortex and for the delays incurred during subsequent cortical processing (*Orban et al., 1985*; *Berry et al., 1999*; *Jancke et al., 2004*; *Subramaniyan et al., 2018*; *Sundberg et al., 2006*).

For example, neurophysiological recordings in animals reveal motion extrapolation mechanisms as early as the retina (*Berry et al., 1999*; *Chen et al., 2013*; *Johnston and Lagnado, 2015*; *Liu et al., 2021*; *Souihel and Cessac, 2021*). By responding to the leading edge of moving stimuli, retinal ganglion cells extrapolate the represented position of those stimuli, and are thought to transmit these extrapolated representations to visual cortex, thereby compensating for some of the lag that arises during transmission. These pre-cortical extrapolation mechanisms should effectively allow primary visual cortex to represent the position of a moving object with reduced latency, as observed in both cat and macaque V1 (*Jancke et al., 2004*; *Subramaniyan et al., 2018*). The existence of these extrapolation mechanisms opens the possibility that transmission delays on the way to visual cortex might

be partially or fully compensated, allowing the early visual system to represent moving objects on predictable trajectories closer to their real-time locations.

It is unclear whether similar mechanisms operate along the cortical visual processing hierarchy to compensate for additional delays that accumulate as visual information is processed. On the one hand, there is suggestive evidence that position representations in areas V4 (*Sundberg et al., 2006*) and V5 (*Maus et al., 2013a*) are shifted for moving objects, potentially reflecting the effect of motion extrapolation in those areas. That interpretation is consistent with recent fMRI (*Schneider et al., 2019*; *Harvey and Dumoulin, 2016*), theoretical (*Hogendoorn and Burkitt, 2019*) and psychophysical (*van Heusden et al., 2019*) work suggesting that motion extrapolation mechanisms operate at multiple levels of the visual system. On the other hand, shifted position representations in higher areas might simply result from those areas inheriting extrapolated information from upstream areas such as V1. To our knowledge, no study to-date has investigated how the represented position of a smoothly moving object evolves over time as visual information about that object flows along the visual hierarchy.

Here, we address this question by using time-resolved EEG decoding to probe the position representations of smoothly moving objects across all levels of the human visual system in real time. We show that early position representations of moving objects are in close alignment with the veridical position of the object, providing the first direct evidence in humans that extrapolation processes allow the early visual system to localise moving objects in real time. We further show that during the course of cortical visual processing, position representations increasingly lag behind real-time stimulus position as information progresses through the visual hierarchy. This suggests that delay compensation is primarily achieved at very early stages of stimulus processing, and that subsequent cortical visual areas do not implement further compensation for neural delays. Nevertheless, this early compensation ensures that the represented position of a moving object throughout the entire visual hierarchy is far more up-to-date than could be expected on the basis of the latencies of neural responses to static objects. These findings demonstrate the existence of significant predictive processing during motion perception, but constrain any predictive mechanisms to acting relatively early in processing.

## Results

Twelve observers viewed sequences of black discs that were either flashed in one of 37 possible positions on a hexagonal grid (static trials) or smoothly moved through a series of positions on the grid

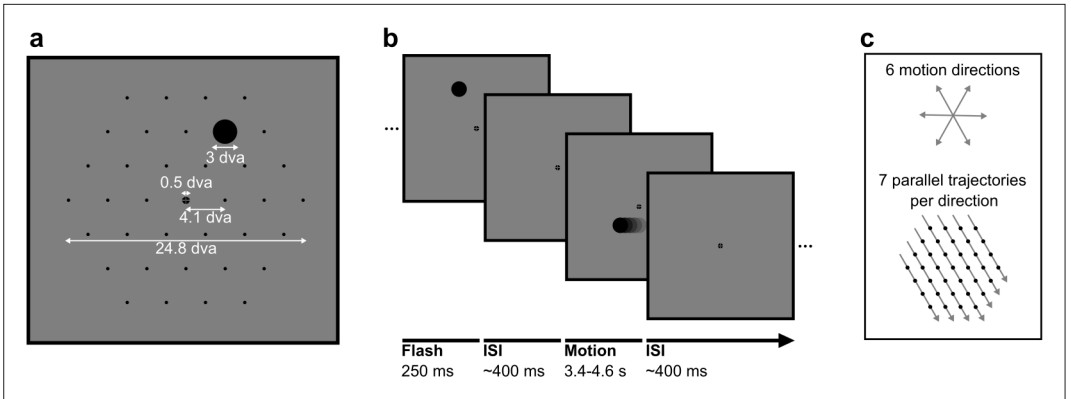

**Figure 1.** Stimuli in static and motion trials. (**a**) Stimulus configuration. Stimuli were presented in a hexagonal grid. In static trials, a black circle was shown centred in 1 of the 37 positions (marked by black dots, not visible during the experiment). In motion trials, the same stimulus moved at 10.36 degrees visual angle/second (dva/s) in a straight line through the grid. A fixation point was presented in the centre of the screen and the background was 50% grey. All measurements are in dva. (**b**) Trial structure. A trial consisted of a black circle flashed in one position for 250 ms (static trials) or moving in a straight line for between 3350 and 4550 ms (motion trials). Trials were randomly shuffled and presented separated by an inter-stimulus interval randomly selected from a uniform distribution between 350 and 450 ms. (**c**) Motion trials. The moving stimulus travelled along 1 of 42 possible straight trajectories through the grid: six possible stimulus directions along the hexagonal grid axes with seven parallel trajectories for each direction. The moving stimulus passed through four to seven flash locations, depending on the eccentricity of the trajectory.

along a straight trajectory (motion trials, *Figure 1*). Static trials were presented 252 times per position, and each of the 42 motion vectors was presented 108 times. EEG data were recorded over six testing sessions and analysed offline (see Methods). Multivariate pattern classifiers (linear discriminant analysis) were trained to discriminate stimulus position for all pairwise combinations of positions, using EEG activity evoked by static stimuli in those positions. These classifiers were subsequently tested on EEG data recorded during an independent subset of static trials, or during motion trials. Results of this classification analysis were combined to estimate the likelihood of the stimulus being present in each of the possible stimulus positions, $p_s(i)$ for $i \in \{1, 2, \ldots 37\}$, where $s$ is the presented position. From this, we traced the evolution over time of the estimated likelihood of the stimulus being present in the position in which it was actually presented (static trials) or moved through (motion trials), $p_s(s)$, hereafter referred to as the *stimulus-position likelihood*.

This analysis was repeated for multiple combinations of training timepoint (i.e. time after onset of a static stimulus) and test timepoint (i.e. time after onset of a static stimulus or within a motion vector). Using different training timepoints allowed us to probe neural representations at different levels of the visual hierarchy, and testing at multiple timepoints allowed us to characterise how information flows through those levels over time during the epoch of interest (*King and Dehaene, 2014*). In this way, we were able to evaluate whether the neural position representation of a moving object flows through the visual hierarchy at the same latency as the position representation of a static flash. Additionally, this allowed us to evaluate how much the position representation of the moving object lags behind that object's physical position.

## Decoding position of static stimuli

First, we investigated the ability of classifiers to discriminate the presented position of static flashes based on the EEG signal. *Figure 2a* shows average classification accuracy across pairwise combinations of positions over time, grouped by distance between the two positions. Classifiers were trained and tested using subsets of data from the same timepoints. As expected, the performance of pairwise classifiers improved with increasing stimulus separation. This is due to the retinotopic organisation of visual cortex; stimuli elicit more distinct patterns of activity when they are further apart.

Pairwise classification results were combined to calculate the stimulus-position likelihood. Stimulus-position likelihoods by eccentricity (distance from fixation) are plotted in *Figure 2—figure supplement 1*. We then averaged across all stimulus positions and participants. This likelihood was compared to a permuted null distribution to establish whether it was significantly above chance at each timepoint (*Figure 2b*, see Methods). The stimulus-position likelihood was above chance starting at 58 ms after stimulus onset, meaning that stimulus positions were decodable from this timepoint onward. An equivalent analysis was applied to eye position during static trials to ensure that EEG decoding results were not confounded by systematic microsaccades (*Figure 2—figure supplement 1*), either due to neural activity related to motor planning, movement-related EEG artefacts, or shifts in retinotopy (*Mostert et al., 2018*). For eye position, stimulus-position likelihood was not significantly above chance at any timepoint, which is in line with previous findings (*Blom et al., 2020*; *Salti et al., 2015*; *Tse et al., 2002*). Therefore, we can conclude that the classifiers trained on the EEG response to the static stimulus did not exploit eye movements to determine the stimulus position.

To assess whether position-related information was stable or variable across the time-course of the visual evoked response, the classification analysis was generalised across time (*King and Dehaene, 2014*): classifiers were trained and tested at all combinations of timepoints. *Figure 2c* shows the resulting temporal generalisation matrix (TGM), averaged across all stimulus positions and participants. The TGM was typical of position decoding plots seen in previous work (*Hogendoorn and Burkitt, 2018*). It appears that the stimulus representations before 150 ms training/test time did not generalise to other timepoints, but later reactivation of representations is apparent after 150 ms (off-diagonal red blobs). Finally, *Figure 2d* shows topographic maps of activation which contributed to classification of stimulus position (*Haufe et al., 2014*); these show that the relevant signal was mainly recorded from occipital electrodes, suggesting a source within the visual cortex, as expected.

## Decoding position of moving stimuli

To decode the position of moving stimuli, we again trained classifiers on pairwise combinations of static stimuli, then applied these classifiers to EEG data recorded during motion trials. An illustration

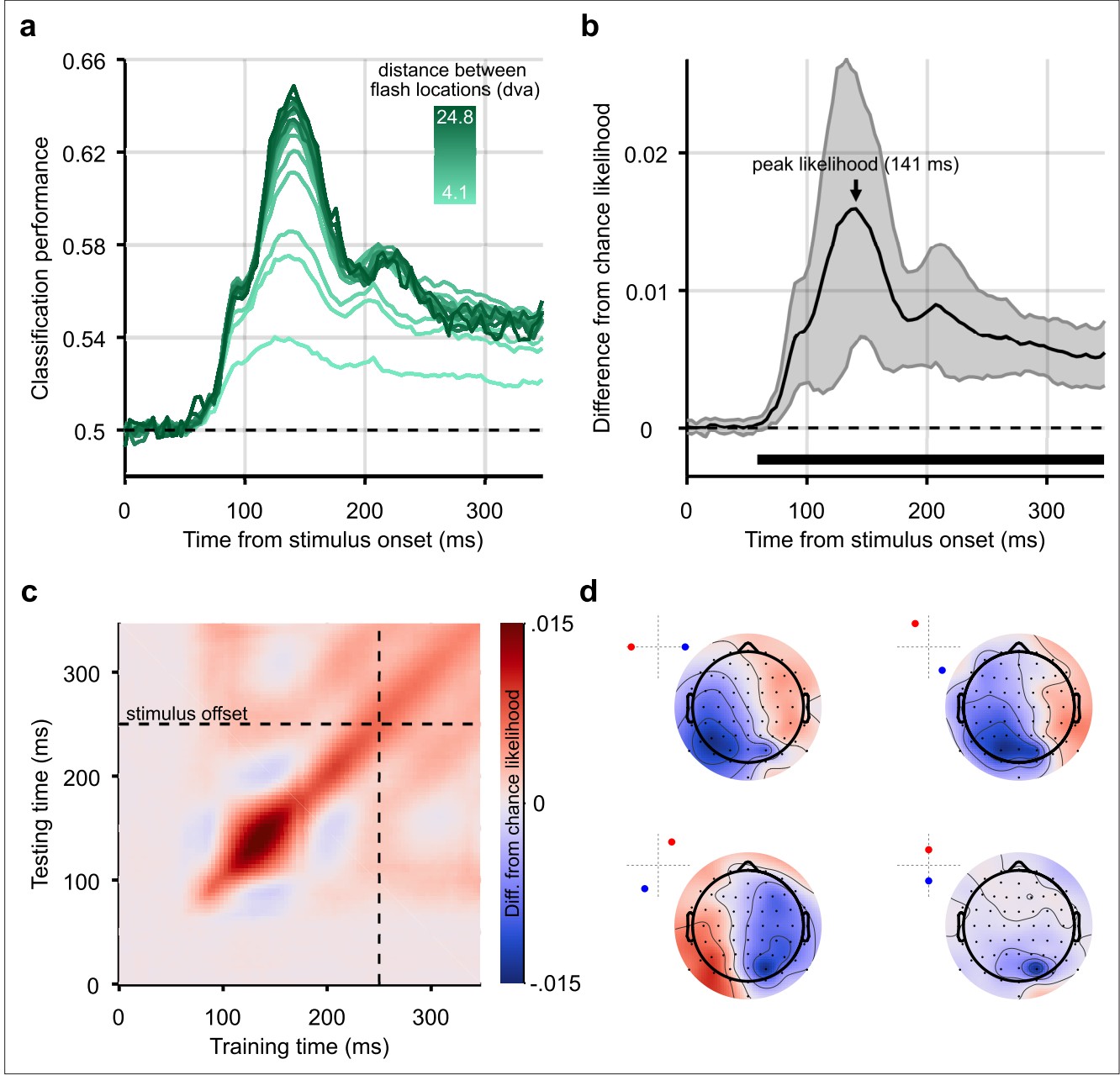

**Figure 2.** Classification results for decoding the position of static stimuli. (**a**) Group-level pairwise classification performance of static stimulus-position discrimination sorted by distance between stimulus positions (separate lines). Classifiers were trained and tested on matched timepoints from 0 to 350 ms (i.e. the time diagonal). (**b**) Timepoints along the time diagonal at which likelihood of the stimulus being in the presented position (stimulus-position likelihood) is significantly above chance (*n*=12, p<0.05, cluster-based correction) are marked by the bar above the x-axis. The stimulus-position likelihood was significantly above chance from 58 ms onward. Shaded error bars show one standard deviation around the mean across observers. Chance level has been subtracted from all likelihoods to demonstrate the divergence from chance, in this graph and all others showing stimulus-position likelihood. (**c**) Stimulus-position likelihood (colour bar; *Kovesi, 2015*) was calculated from classification results at each combination of training and test times. Results averaged across all stimulus positions and participants are displayed as a temporal generalisation matrix (TGM). (**d**) Topographic maps show participant-averaged topographic activity patterns used by classifiers to distinguish stimulus positions at 141 ms post stimulus onset, the time of peak decoding (marked by an arrow in panel b). Insets in the top left of each scalp map show which two stimulus positions the classifier has been trained to discriminate. Scalp maps were obtained by combining classification weights with the relevant covariance matrix. As expected, for all four comparisons, activation was predominantly occipital and, when the stimulus positions were on either side of the vertical meridian, lateralised.

The online version of this article includes the following figure supplement(s) for figure 2:

**Figure supplement 1.** Stimulus-position likelihood as a function of eccentricity.

**Figure supplement 2.** Analysis of eye movements.

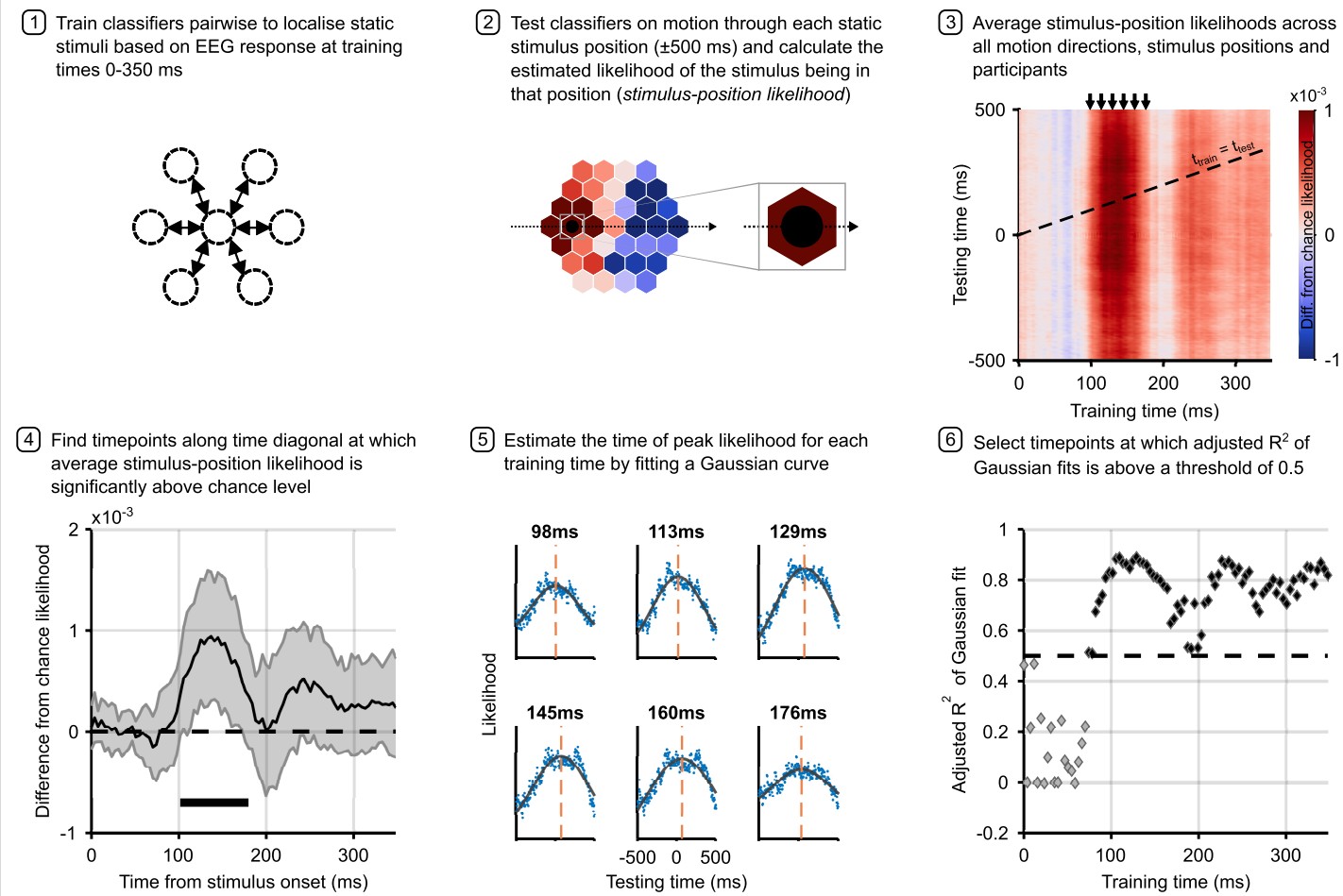

**Figure 3.** Analysis pipeline for motion trials. Panels describe steps in calculating the time to peak stimulus-position likelihood in motion trials, including graphs of relevant data for each step. Steps 1 and 2 describe the classification analysis applied to obtain the stimulus-position likelihoods. The figure in step 3 shows the group-level temporal generalisation matrix for training on static stimuli and testing on moving stimuli. The black dotted line shows the 'diagonal' timepoints, where the time elapsed since the moving stimulus was at the flash position equals the training time. Step 4 shows timepoints along this diagonal at which the stimulus-position likelihood was significantly above chance, as established through permutation testing. Significance is marked by the solid black line above the x-axis; the likelihood is significantly above chance from 102 to 180 ms ($n=12$, $p<0.05$, cluster-based correction). Shaded error bars show one standard deviation around the mean. The figure in step 5 shows the same data as step 3 for selected training times (arrows above temporal generalisation matrix [TGM] correspond to subplot titles). Each subplot shows a vertical slice of the TGM. Blue points show data, to which we fit a Gaussian curve (black lines) to estimate the time of peak likelihood for each training time (dashed orange lines). These are the data points plotted in **Figure 4a and b**. Step 6 shows adjusted $R^2$ of Gaussian fits for each training timepoint. A cutoff of 0.5 was used to select timepoints at which the Gaussian fit meaningfully explains the pattern of data.

The online version of this article includes the following figure supplement(s) for figure 3:

**Figure supplement 1.** Fitted Gaussian curves at extended training times.

of each step in the analysis of motion trials is shown in **Figure 3**. As before, the stimulus-position likelihood was calculated, this time at each timepoint during motion epochs.

We considered EEG epochs from 500 ms before to 500 ms after the timepoint at which the moving stimulus was exactly in each possible static stimulus location. This time-window was chosen to be broad enough to capture stimulus-evoked activity as the stimulus approached and receded from each position (moving from one position to the next took 400 ms). We then averaged the time-course of stimulus-position likelihoods across all six motion directions and 37 stimulus positions. The first position along each trajectory was excluded due to observed strong EEG responses to the initial onset of the stimulus.

The TGM derived from classifiers trained on static trials and tested on motion trials (*Figure 3*, step 3) revealed that classifiers trained on timepoints from around 100 ms were able to decode the position of moving objects. To identify timepoints at which classification was significantly above chance, we considered the performance of classifiers trained and tested on matching timepoints (diagonal of the TGM). Permutation testing revealed that decoding was significantly above chance for timepoints between 102 and 180 ms (*Figure 3*, step 4). Note that because we are investigating possible latency differences between the neural response to static and moving stimuli, maximal decoding is likely achieved off-diagonal, making this a conservative analysis choice.

Although the average stimulus-position likelihood was smaller in magnitude for moving stimuli compared to static stimuli, we observed that the location-specific neural response to motion over time was characterised by a gradual increase of the stimulus-position likelihood as the stimulus approached the centre of the position, then a decrease as the stimulus moved away on the other side. This is illustrated in *Figure 3*, step 5 (see *Figure 3—figure supplement 1* for more training timepoints), and is similar to the pattern of activity found in response to a moving bar with direct recordings from cat V1 (*Jancke et al., 2004*).

## Establishing the latency of position representations of moving stimuli

To investigate the latency at which neural position representations are activated for moving objects, we calculated the timepoint at which the peak stimulus-position likelihood was reached during motion sequences. Again, this was repeated for different training times as a proxy for different stages of neural processing. The time to peak likelihood in the test data for each training timepoint reflects the time at which the location-specific activity is most similar in the train and test set. This is assumed to be the time that the brain is representing the moving stimulus at the centre of a flash position. We use *peak* likelihood, as opposed to onset or a peak percentage, as the latency measure due to possible variations in receptive field (RF) size over the course of visual processing. As time elapses during stimulus processing, visual information reaches visual areas further up the processing hierarchy, which contain stimulus-selective neurons with larger RFs (*Johnson et al., 2021*; *Harvey and Dumoulin, 2011*). This means that a moving stimulus would enter the RF earlier in these later-activated brain regions. Looking at the peak neural response avoids this problem, because peak response would be expected when the stimulus is at the centre of the RF, irrespective of RF size.

To establish the latency with which the position of a moving object is represented at different stages of visual processing, we identified the timepoint at which our classification analysis yielded maximum stimulus-position likelihood. This was achieved by fitting a Gaussian curve to the observed time-course of the calculated likelihood averaged across participants, separately for each training time (*Figure 3*, step 5). Adjusted $R^2$ of these fits can be found in *Figure 3*, step 6. For training timepoints later than ~80 ms, the Gaussian curves provided a very good fit to the evolution of stimulus-position likelihood over time, with $R^2$ values over 0.5. Although the window of significant cross-classification of static stimuli to moving stimuli is restricted (*Figure 3*, step 4), the sustained high adjusted $R^2$ values indicate that even for training times at which the stimulus-position likelihood was close to chance level, the likelihood increased and decreased as the stimulus traversed each flash location.

*Figure 4a* shows the time to peak likelihood for motion across all training timepoints at which adjusted $R^2$ exceeded a minimum value of 0.5. The choice of $R^2$ cutoff is essentially arbitrary, but the pattern of points in *Figure 3*, step 6 shows that this selection is relatively robust to changes in the cutoff value. Up to ~150 ms training time, the time to peak likelihood increases with increasing training time. This follows the same pattern as the static trials (see *Figure 4b*), where earlier representations of the stimulus (i.e. early training times) were activated at a shorter latency in the testing epoch than later representations of the stimulus. This sequential pattern is consistent with the first feedforward sweep of stimulus-driven activation. As information flows through the visual processing hierarchy, representations of the stimulus will gradually change over time. The order of these changes appeared to be consistent between static and motion trials.

This pattern subsequently reverses between 150 and 200 ms, indicating that hierarchically later representations were activated at a shorter latency. Finally, from a training time of ~250 ms the time to peak likelihood was stable at approximately 50 ms. The non-monotonic relationship between training time and time to peak likelihood could emerge because there was variable compensation for neural delays at different training times. Perhaps more likely, this pattern could reflect feedforward and

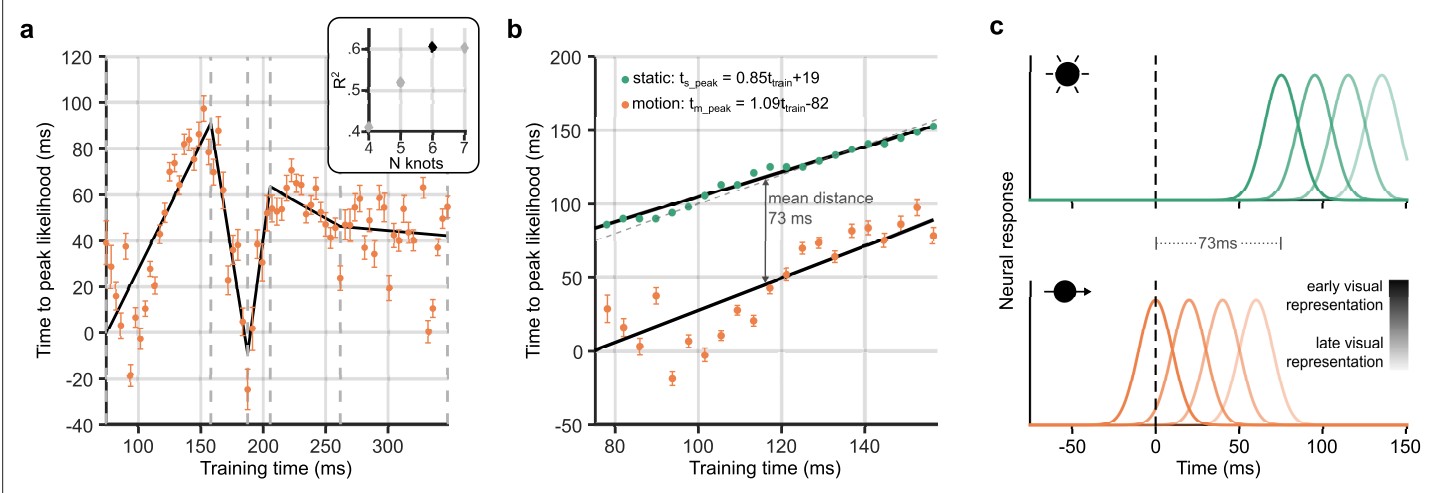

**Figure 4.** Neural response latency during motion processing. (**a**) Latencies of the peak stimulus-position likelihood values during motion processing. The timepoint at which peak likelihood was reached is plotted against training time. Error bars around points show bootstrapped 95% confidence intervals of the peak shift parameter of the Gaussian fit, computed from $n = 12$ participants (see **Figure 3**, step 5). It can be observed that the peak time increases and decreases, then levels out. Points of inflection within this timeseries were identified using piecewise regression (shown in black). The number of inflection points, or knots, was established by comparing the $R^2$ of piecewise regression fits, as shown in the inset graph. It was determined that six knots was optimal; positions of these knots are marked by grey dotted lines on the main graph. (**b**) Time to peak likelihood during the initial feedforward sweep of activity through the visual cortex. Displayed is a subset of points from those shown in panel a, corresponding to a restricted time-window between the first two knots, during which the first feedforward sweep of activity was most likely occurring. The dotted diagonal shows the 45° line, where the time of peak likelihood would equal the training time. Data points from static trials (green) should theoretically lie along this line, as, in this case, the training and test data were subsets of the same trials. Straight lines were fit separately for static ($F(21,19) = 805.45$, $p=1.24 \times 10^{-17}$) and motion trials ($F(21,19) = 40.91$, $p=3.07 \times 10^{-6}$). Both lines had similar gradients, close to unity, indicating equivalent cumulative processing delays for static and motion trials within this training time-window. However, the intercept for motion was much earlier at –80 ms. The mean distance between the two lines is marked, indicating that position representations were activated ~70 ms earlier in response to a moving stimulus compared to a flashed one in the same location. Time to peak likelihood at the beginning of the feedforward sweep was approximately 0 ms, indicating near-perfect temporal alignment with the physical position of the stimulus. (**c**) Illustration of compensation for neural delays at different cortical processing levels. The static stimulus (green, top) and the moving stimulus (orange, bottom) are in the same position at time = 0 ms (black dashed line), but there is a 73 ms latency advantage for the neural representations of the stimulus in this position when it is moving. Each separate curve represents neural responses emerging at different times during stimulus processing, where higher contrast corresponds to early visual representations and lower contrast corresponds to later visual representations. The earliest neural response, likely originating in early visual cortex, represents the moving stimulus in its real-time location. The consequence of this is a shift in the spatial encoding of the moving stimulus: by the time neural representations of the flash emerge, the moving stimulus will be represented in a new location further along its motion trajectory. The relative distance between the subsequent curves is the same for moving and static stimuli, because there is no further compensation for delays during subsequent cortical processing.

feedback sweeps of activity in the visual cortex evoked by the static stimulus: the feedforward sweep activates sequential representations, and information flowing backward along the hierarchy reactivates the same activity patterns in reverse order. The timescale of this wave of activity was in line with previous findings from **Dijkstra et al., 2020**, who showed approximately 10 Hz oscillations evoked by face/house stimuli. Additionally, previous TMS, MEG, and fMRI results suggest that 150 ms is a reasonable estimate for the time it takes for visual information to reach hierarchically later visual processing areas, such as V5/MT+ (**Sack et al., 2006**; **Ahlfors et al., 1999**; **Hotson et al., 1994**; **Yoshor et al., 2007**; **Mohsenzadeh et al., 2018**; **Lamme and Roelfsema, 2000**). This would imply that the activity evoked by static compared to moving stimuli is different after about 150 ms, due to a divergence in further stimulus processing. Therefore, cross-generalisation of position decoding across stimulus types is possible after this timepoint because a portion of the flash-evoked signal reflects reactivation of earlier activity patterns, which are common to static and moving stimulus processing. If this later activity (>150 ms training time) does indeed reflect feedback processing of the static stimulus, then, for these later timepoints, the latency measure we have calculated might not be informative about the time necessary to first represent the moving object, because the initial activations and the reactivations are indistinguishable.

In order to confirm which timepoints predominantly correspond to feedforward processes rather than feedback, we applied piecewise regression, as implemented in the Shape Language Modeling

toolbox (*D'Errico, 2022*). In this approach, several polynomials are smoothly joined together at 'knots'. Placement of knots, at the start and end of each segment, is optimised by reducing root mean squared error. We fit straight lines, and varied the number of knots between four and seven, in order to identify the optimal number. The best piecewise regression fit, with six knots, is shown in *Figure 4a*. Corroborating our observation, the first internal knot was placed at 158 ms. We took this inflection point as the end of the initial feedforward sweep of information through the visual cortex. The piecewise regression revealed further positive and negative slopes, suggesting that feedforward and feedback sweeps of activity continue during later stimulus processing (*Dijkstra et al., 2020*).

## Latency advantages for moving stimuli during feedforward processing

Having identified the timepoints during the motion epochs corresponding to early feedforward processing, we further investigated the relationship between training time and the time of peak stimulus-position likelihood, to compare the latency with which the brain represents static and moving stimuli. *Figure 4b* shows the peak time for moving objects during the feedforward sweep, along with the first segment of the fitted piecewise linear regression. We were interested in comparing the latency and time-course of stimulus-related processing of static and motion trials in this restricted time-window, to establish whether the position of moving objects was predictively encoded. We therefore established time to peak likelihood for these same training timepoints in the static trials. Because the participant-averaged time-course of the stimulus-position likelihood for each training time was much less noisy for static than motion trials, the time of peak likelihood was computed as a simple maximum for each training time. Qualitatively, it can be observed that this lay along the diagonal of the TGM (*Figure 2b*, green points).

For the static stimuli, a linear fit relating training time and time to peak likelihood (*Figure 4b*, upper line) was very similar to the 45° line (*Figure 4b*, grey dashed line), revealing that each representation of the stimulus was active at roughly the same time in the training and test data. A linear fit to the static data points was significantly better than a constant model ($F(21,19) = 805.45$, p=$1.24 \times 10^{-17}$). The line which best described the relationship between static time to peak likelihood and training time was:

$$t_{\text{s\_peak}} \sim 0.85 t_{\text{train}} + 19. \tag{1}$$

Both the intercept and gradient parameters were significantly different from zero ($t$=5.48, p=$2.32 \times 10^{-5}$; $t$=28.38, p=$1.24 \times 10^{-17}$). The line had a gradient close to one (95% CIs: 0.79–0.92) and a small intercept (95% CIs: 12–27 ms), indicating only a small shift in the peak time between training and testing. This line fit indicates that the patterns of activity on which the classifiers were trained were most similar to activity at approximately the same timepoint in the test trials. This is as expected, as train and test data are subsets of the same static trials.

In contrast, we found evidence of a shift in the latency of activation of representations of the moving stimulus. A regression line was computed also for time to peak likelihood in the motion epochs ($F(21,19) = 40.91$, p=$3.07 \times 10^{-6}$), with equation:

$$t_{\text{m\_peak}} \sim 1.09 t_{\text{train}} - 82. \tag{2}$$

Both parameters were again significantly different from zero (intercept: $t$=–4.06, p=$6.12 \times 10^{-4}$, 95% CIs: 95% CIs: –124 to –40 ms; gradient: $t$=6.40, p=$3.07 \times 10^{-6}$, 95% CIs: 0.73–1.45). Furthermore, the difference between the static and motion gradients was not significantly different from zero ($t$=–1.58, p=0.13) but the difference between the two intercepts was ($t$=5.32, p=$3.32 \times 10^{-5}$).

This lack of difference between the gradients means that, once position information was available in the cortex, successive cortical representations were sequentially activated along the same time-course for moving and static stimuli. In other words, delays that accumulate during cortical processing did not appear to be compensated when processing motion. This is illustrated in *Figure 4c*, where the relative delay between neural representations is preserved regardless of whether the stimulus is moving or not. Importantly, however, the static and motion intercepts were significantly different. The linear fit to time to peak likelihood for motion stimuli had a large negative intercept of –82 ms (*Equation 2*), which was substantially lower than the intercept for static trials (19 ms, *Equation 1*). At the beginning of the time-window of interest ($t_{\text{train}}$ = 75 ms), the motion regression line crossed the y-axis at –1 ms, while the static regression line crossed at 83 ms (see *Figure 4b*).

To summarise, the lack of difference in the slopes of the regression lines indicates that, once early representations of position are activated, similar delays are incurred as position-related information flows through the visual cortical hierarchy. The observed shift in the intercept signifies that stimulus positions are represented much earlier for moving compared to static stimuli.

The mean distance between the two lines was calculated at 73 ms, implying that the position of a moving object was represented with a latency that was approximately 70 ms shorter than a static object in the same position. For early neural position representations (training times around 70–80 ms), the latency of the peak in the position representation was approximately 0ms. In turn, this means that these neural representations were activated at the time that the moving object was physically centred on the corresponding position. This can also be seen in *Figure 4c*: the representations of the moving stimulus emerge 73 ms earlier than the same representations of the flashed stimulus. The earliest representation of the moving stimulus peaks when $t=0$ ms, the time at which the moving stimulus is in the same position as the flash. This corresponds to a spatial shift in the encoding of the moving stimulus, as, by the time the representations of the flash emerge, the moving stimulus will be represented further ahead on its path. Based on the training time, these early stimulus representations likely originated in early visual cortex (V1-3), meaning that the early visual system was able to almost completely compensate for neural delays accumulated during processing up to that point and represent moving objects close to their real-time position.

## Discussion

In this study we investigated how the visual system compensates for neural transmission delays when tracking the positions of moving objects. We investigated the latency of neural position representations of moving stimuli compared to unpredictably presented static stimuli, as well as the real-time position of the stimulus. By computing the timepoint at which each position representation of the moving stimulus was most similar to a static stimulus in the same location, we tracked the represented position of the stimulus over time across the visual hierarchy.

We demonstrate that classifiers trained to locate static stimuli based on the stimulus-evoked EEG signal could also localise moving stimuli. This is the first study, to our knowledge, to demonstrate cross-classification between stationary and smoothly moving stimuli with EEG, showing that population neural codes for location are (at least partially) shared between the two stimulus types. We, therefore, were able to investigate the timing of neural responses to motion in humans with a fine temporal resolution. We subsequently showed that, during the first feedforward sweep of activity, the neural response encoding the position of a moving stimulus was shifted approximately 75 ms earlier than the response to a static stimulus. The early decoded representations of the position of a moving stimulus aligned with the real-time position of the object, rather than the position corresponding to afferent retinal input (subject to transmission and processing delays), which would instead signal outdated position information in visual cortex. Finally, we showed that delay compensation was primarily achieved before information reached visual cortex, as the recorded cortical processing of static and motion stimuli followed a similar time-course. Overall, this study shows direct neural evidence of motion extrapolation enabling accurate real-time representation of moving objects in humans.

These results are consistent with findings of RF shifts across the visual cortex in response to motion. Many earlier fMRI studies showing RF shifts against the direction of motion *Liu et al., 2006*; *Whitney et al., 2003*; *Raemaekers et al., 2009* have been dismissed because of the 'aperture-inward' bias, in which the trailing edge of a motion stimulus evokes larger responses than the leading edge (*Wang et al., 2014*). This is not an issue for the present study, as we can determine the timing of neural responses at a fine temporal scale, rather than looking at aggregate responses over whole motion trajectories. Neural recordings from animals and more recent fMRI studies in humans have reliably shown RF shifts throughout the visual cortex in response to motion (*Harvey and Dumoulin, 2016*), and that these displacements are against the direction of motion (*Schneider et al., 2019*; *Sundberg et al., 2006*; *Fu et al., 2004*). However, several differences remain between the previous fMRI results and the present study. *Harvey and Dumoulin, 2016*, found that RF shifts in response to motion scale with the size of the RF across the visual hierarchy. This implies that visual areas higher up the processing hierarchy that are activated later in time would shift their RFs more than lower visual areas which are activated earlier. In contrast, our results suggest that later visual areas show RF shifts of

the same magnitude as earlier visual areas. However, it is not clear whether it is possible to map time elapsing after stimulus onset in EEG to processing in different visual areas as recorded by fMRI. The longer timescale of the fMRI signal means that it could be indexing later activity than we are recording with EEG, or include signals that emerge after integration of many feedforward and feedback sweeps of activity. While further research is needed to understand how extrapolation operates at different spatial scales, this converging evidence of RF shifts against the direction of motion suggests that the positions of moving objects are predictively encoded during processing. Furthermore, we provide evidence that RF shifts occur during the initial feedforward sweep of the visual response.

## Mechanisms of compensation for delays

Our findings point to several mechanisms that have been proposed to compensate for neural delays. We found that the early visual response to moving stimuli is shifted in time, such that the neural delays accumulated up to that point are compensated. However, during subsequent cortical processing, there is no further compensation for delays. As discussed in the Introduction, retinal ganglion cells respond strongly to the leading edge of moving stimuli (*Berry et al., 1999*; *Johnston and Lagnado, 2015*). This effectively shifts the encoding of the position of a moving stimulus forward relative to a static stimulus at the earliest stage of processing. Additionally, evidence of a latency advantage for moving stimuli has been identified in the cat lateral geniculate nucleus of the thalamus (*Orban et al., 1985*), where visual information is transmitted en route to the visual cortex. However, none of the previous evidence suggests that these pre-cortical mechanisms are sufficient to account for compensation for neural delays, to the extent we observe here. In particular, the retinal compensation mechanisms only appear to act up to speeds of about 5 degrees visual angle/second (dva/s) (*Berry et al., 1999*; *Jancke et al., 2004*), roughly half the speed used in the current study. This is thought to be achieved by a large response to the leading edge of the stimulus, followed by gain control mechanisms reducing neuronal firing rates. In the present study, it takes 400 ms for the stimulus to travel from one static stimulus position to the next, so the leading edge of the stimulus is closest to the stimulus position it is approaching 200 ms before it reaches that position. In general, we see a ramping in the likelihood earlier than –200 ms (see *Figure 3—figure supplement 1*), so we believe that this adaptation is unlikely to be the sole mechanism responsible for the observed shift in encoded location.

Therefore, it is likely that some cortical mechanisms play a role. For example, there is evidence that a model of object motion is encoded in MT+, and influences neural response profiles in earlier visual areas through feedback connections (*Maus et al., 2013a*; *Maus et al., 2013b*; *Nijhawan and Kirschfeld, 2003*). These feedback connections could transmit information to neurons into whose RFs the moving stimulus will soon enter, driving an anticipatory response. Similarly, within-layer horizontal connections might activate neurons further ahead on the motion path (*Benvenuti et al., 2020*). *Benvenuti et al., 2020*, show that this input from feedback and horizontal connections can drive spiking responses in cat V1. If information about object motion is used to compensate for delays via feedback connections, it could be expected that the particular speed of the stimulus would be exploited to maximise the accuracy of the position representation. In the present study, we only tested one stimulus speed, so further research including a range of speeds, as found in natural scenes (*Wojtach et al., 2008*), is necessary to examine whether compensatory mechanisms work in this way. Crucially, our findings suggest that these mechanisms act *only* early in the course of stimulus processing, and therefore are present only early in the visual cortical hierarchy.

Even though we find temporal alignment between the early representations of the stimulus and its physical position, this alignment is lost during further processing. In a recent theoretical paper, *Hogendoorn and Burkitt, 2019*, argue that cortical motion extrapolation is necessary to minimise the discrepancy (prediction error) between an internal model of object position and the external world in the case of time-varying stimulation. Proposed are two possible implementations of this cortical extrapolation: either delays are compensated through extrapolation in both feedforward and feedback activity, or, alternatively, extrapolation only occurs in feedback activity. Although the authors claim that the model including feedforward and feedback extrapolation is more parsimonious, this study suggests that feedforward cortical delays are not compensated. We therefore support the proposition that, if prediction errors are to be minimised, extrapolation might be implemented only in feedback connections. However, the present analysis approach may not be suitable to uncover this process, as cortical extrapolation could be a motion-specific computation enacted by different neural

populations from those that encode static stimuli. Although this analysis captured feedforward and feedback sweeps of activity through the visual cortex, we believe that these oscillations are present in the processing of the flash, not the motion. This would mean that the portion of the later flash-evoked signal that that cross-generalises to motion reflects reactivation of earlier stimulus representations rather than further stimulus processing. However, there remains the possibility that these oscillations are evoked by motion. This would imply that, during feedback activity, the position of the stimulus is not updated as the stimulus moves, and that the stimulus is represented in multiple locations concurrently in the visual system (*King and Wyart, 2021*). Further research is needed to tease apart these options. Nevertheless, a complete model of compensation for neural delays in motion perception should account for extensive extrapolation early in visual processing, as observed here.

## Comparison between smooth and apparent motion

Previous EEG research has investigated whether neural delays are compensated when viewing apparent motion. Apparent motion is a visual illusion in which stimuli that appear sequentially along a motion path, but are temporally and spatially separated, are perceived as a single stimulus in motion. Although apparent motion is an impoverished motion signal, two studies have found evidence of a 15–20 ms latency advantage when processing a stimulus within an apparent motion sequence compared to an unpredictable flash (*Hogendoorn and Burkitt, 2018*; *Blom et al., 2021*). This is substantially less compensation than we find in the present study, likely due to the lack of continuous stimulation in the case of apparent motion. For example, an apparent motion stimulus is static on the retina, precluding the gain control mechanism described above from acting. Smoothly moving objects also trigger a travelling wave of activity that propagates in the visual cortex in front of the retinotopic position of the stimulus (*Benvenuti et al., 2020*), which is different from the non-linear combination of activity elicited by an apparent motion stimulus (*Chemla et al., 2019*). As a lot of natural motion stimuli provide continuous input and, therefore, recruit the entire motion-sensitive visual pathway, it is important to characterise how smooth motion is processed. *Blom et al., 2020*, additionally found evidence that a sensory template of an expected apparent motion stimulus is activated before the onset of the stimulus, which has additionally been found in monkey V1 (*Guo et al., 2007*). An equivalent preactivation could be present in the case of smooth motion, but in the current study, processing of the presented stimulus would obscure any predictive activity. *Blom et al., 2020*, also found that when the stimulus reversed direction unexpectedly, it was briefly represented in the expected next position, and that there is a latency disadvantage associated with the first unexpected stimulus after a reversal (*Blom et al., 2021*); future research should extend the present findings by investigating the neural consequences of an unexpected change of direction, and generally unpredictable motion, in a smooth motion trajectory. Delays seem to be compensated to a greater extent in this case of smooth motion, so the visual system would have to employ a larger correction for erroneous position encoding if the stimulus changed direction.

## Limitations

A limitation of the present study is that the localisation accuracy of moving stimuli was considerably lower than that of static stimuli. This is because classifiers were trained and tested on different stimulus types; neural populations that encode the position of static stimuli do not completely overlap with neural populations that encode the position of moving objects (*Noda et al., 1971*; *Subramaniyan et al., 2018*). Additionally, previous fMRI studies show that, following a strong onset response, the neural response to moving stimuli decreases over time (*Schellekens et al., 2016*; *Schellekens et al., 2017*; *Harrison et al., 2007*; *McKeefry et al., 1997*). This potentially leads to a decreasing signal-to-noise ratio over the course of a single motion trial. Furthermore, because motion epochs were quite long (up to 5 s), the later parts of each motion trial could have been susceptible to slow drift of the EEG signal. Nevertheless, significant cross-classification between static and motion trials was still achieved, ruling this out as a major problem.

Because of the spatial uncertainty associated with EEG, we do not know exactly where signals originate in the brain; source localisation in EEG is an ill-posed problem without co-registration with fMRI (*Jatoi et al., 2014*). While the initial response is likely to be dominated by a feedforward cascade through the visual hierarchy (*Lamme and Roelfsema, 2000*), it is possible that activity recorded at later timepoints in the training epoch reflects ongoing processing in hierarchically early visual areas, as

well as additional processing in hierarchically later visual areas. However, this issue does not alter our conclusions concerning the relative timing of activity when viewing static or moving stimuli.

We additionally found that the earliest signals containing information about the location of static stimuli were not informative about moving stimuli. The timing of the earliest flash-evoked activity (~60 ms after stimulus onset) suggests a source within V1 (*Alilović et al., 2019*; *Fahrenfort et al., 2007*; *Di Russo et al., 2002*; *Wibral et al., 2009*; *Vanni et al., 2004*). In contrast, the position of the moving stimulus was decodable only on the basis of representations formed after approximately 100 ms, suggesting that this analysis approach does not capture the earliest motion-evoked V1 activity. One possibility is that, due to variability in when stimulus processing begins across trials (*Vidaurre et al., 2019*), the signal-to-noise ratio in the static trials at these earlier timepoints may be too low to cross-generalise to moving stimuli. Alternatively, early processing of motion could be different to static stimuli: there is some evidence that direct connections from either LGN or the pulvinar to MT+ (bypassing V1) are used when processing motion (*ffytche et al., 1995*). This issue is hard to overcome, as training classifiers on moving stimuli would render capturing latency differences impossible; any latency shift in the test data would also be present in the training data. However, one promising approach was taken by *Benvenuti et al., 2020*, who used recordings of monkey V1 to compare responses to trajectories of different lengths. They found that response latency decreased with increasing trajectory length: sub-threshold activation built up in front of the moving stimuli, preparing neural populations to fire upon the arrival of the stimulus in their RF. A similar approach could be taken in human EEG research to avoid the comparison between moving and non-moving stimuli. Additionally, this line of research would benefit from use of fMRI co-registered with EEG, which provides the temporal and spatial resolution necessary to pinpoint signals to a particular time, stimulus position, and neural source.

## The flash-lag effect

Of relevance to these results is the flash-lag effect (FLE), a visual illusion in which a moving bar is perceived ahead of a flashed bar despite them being physically aligned (*Nijhawan, 1994*). This illusion demonstrates that moving objects are indeed perceived in an extrapolated position. Theories of the FLE can largely be sorted into two camps: spatial and temporal explanations (*Maus et al., 2013b*). Spatial models, for example motion extrapolation (*Nijhawan, 1994*; *Hogendoorn, 2020*), suggest that the encoded positions of moving objects are shifted forwards to compensate for neural delays. In contrast, temporal models, for example differential latencies (*Whitney and Murakami, 1998*; *Whitney and Cavanagh, 2000*), suggest that motion is processed faster than flashes or that there is a temporal integration window over which position signals are averaged (*Krekelberg et al., 2000*; *Whitney et al., 2000*). A range of psychophysical evidence has been presented to support each of these theories (and others), suggesting they all play a role in the FLE and, therefore, motion processing. However, our results are congruent only with spatial explanations; temporal models cannot explain how latency shifts could be greater than the latency of the unshifted neural response. We show that parts of the visual system encode moving objects at a position that afferent sensory information could not yet indicate. A similar result was found using EEG analysis of apparent motion (*Blom et al., 2020*), where a sensory template of an expected stimulus within the apparent motion sequence was pre-activated, before any sensory evidence was present. An outstanding question remains about whether neural representations of moving objects flexibly incorporate information about stimulus speed, as seen in animal V1 recordings (*Jancke et al., 2004*; *Subramaniyan et al., 2018*) and the FLE (*Wojtach et al., 2008*).

## Conclusion

This study used multivariate analysis of EEG data to investigate the latency of position representations of moving and static stimuli. We show that, during the first feedforward sweep of activity, the latency of the neural response to moving stimuli is substantially reduced compared to the response to unpredictable static stimuli. The effect of this latency advantage is that early visual areas represent moving objects in their real-time position, suggesting that (potentially a combination of) retinal, subcortical, and cortical extrapolation mechanisms can overcome neural delays very early on in visual processing. Additional delays accumulated during subsequent cortical processing appear not to be compensated. These results demonstrate that the visual system predictively encodes the position of moving stimuli,

and provide an evidence base to constrain models of how and when motion extrapolation is achieved in the human visual system.

## Methods

### Participants

Twelve participants (two male; mean age = 27.0 years, s.d.=4.93 years) completed all six testing sessions and were included in analyses. While no a priori sample size calculation was conducted, we chose to collect a large EEG dataset from relatively few participant to ensure reliable classification of the stimulus. The 12 included participants were drawn from a larger initial pool, including an additional 15 participants that completed only the first session, which was used for screening. Of these additional participants, two withdrew from the study, three were excluded as the eyetracker could not consistently track their eye position, and the remaining ten were excluded after analysis of their first session data, due to poor fixation (more than 15% of trials with fixation lost) or poor EEG classification performance (less than 51.5% average classification accuracy when discriminating the location of static stimuli). Exclusion criteria included requiring glasses to view the computer screen and a personal or family history of epilepsy. Participants were recruited online and gave written informed consent before participation. Participants were reimbursed AU$15 /hr for their time, as well as an additional AU$20 if they completed all six sessions. Ethical approval was granted by the University of Melbourne Human Research Ethics Committee (Ethics ID: 1954628.2).

### Experimental design

Stimuli were presented using MATLAB Version R2018a and the Psychophysics Toolbox extension Version 3 (*Brainard, 1997*; *Pelli, 1997*; *Kleiner et al., 2007*). Stimuli were presented on an ASUS ROG PG258 monitor (ASUS, Taipei, Taiwan) with a resolution of 1920×1080 running at a refresh rate of 200 Hz. Participants were seated, resting their heads on a chinrest, at a viewing distance of 50 cm from the screen in a quiet, dimly-lit room.

*Figure 1* shows the stimulus configuration and trial structure of the experiment. Stimuli were presented on a grey background, with a central fixation target (*Thaler et al., 2013*). Stimuli were black, filled circles with a radius of 1.29 dva presented in a hexagonal configuration with 37 possible stimulus positions. A trial consisted of the stimulus flashing in a single location on the grid for 250 ms (static trials), or moving in a straight line at a velocity of 10.36 dva/s through the grid (motion trials), such that the amount of time spent travelling the length of the stimulus diameter was the same as the duration of the static stimulus. Motion vectors started and finished 7 dva away from the grid to reduce the effects of stimulus onset on the EEG signal. The stimulus passed through between four and seven flash positions, depending on the eccentricity of the vector, taking 400 ms to travel between grid positions. Static and motion trials were randomly shuffled within each experimental session, with an inter-stimulus interval randomly selected from a uniform distribution between 350 and 450 ms. In each testing session, each static stimulus location was repeated 42 times, while each of the 42 motion vectors (6 directions × 7 parallel starting positions; *Figure 1c*) was repeated 18 times. Trials were split into seven blocks, with a duration of approximately 9 min each. After each block, participants could rest and sit back from the chinrest. Six times within each block (every 50 trials), participants could take a mini-break, in which the experiment was paused but they were required to remain in the chinrest. This procedure was repeated over six sessions, totalling 252 static trials in each location and 108 repetitions of each motion vector.

Participants performed a simple target detection task in order to ensure they attended to the stimuli. While maintaining fixation on the fixation point at the centre of the screen, they responded as quickly as possible with the space-bar when the stimulus flashed red for 40 ms. This happened at random 45 times per block, and trials containing a target were discarded from analysis to ensure that the target and response did not interfere with EEG analysis. Each of the target trials was then repeated at the end of the block without a target to maintain equal trial numbers for each static stimulus position/motion vector. Participants completed one practice block of 20 trials at the start of the first session to become acquainted with the task. The practice block could be repeated upon request.

EEG and eyetracking data were collected from participants while they viewed the stimuli. Eyetracking data were collected using an EyeLink 1000 eyetracker (SR Research). The eyetracker was

calibrated at the start of each block, and drift correction was applied after each mini-break. The conversion of the EyeLink 1000.*edf* files to.*mat* files and offline fixation checks were performed with the EyeCatch toolbox (*Bigdely-Shamlo et al., 2013*).

Continuous EEG data were recorded at 2048 Hz using a 64-channel BioSemi Active-Two system (BioSemi, Amsterdam, The Netherlands), connected to a standard 64-electrode EEG cap. Two external electrodes were placed on the mastoids, to be used as a reference. Electrooculography was recorded using six electrodes: on the canthi (horizontal) and above and below the eyes (vertical).

## EEG pre-processing

EEG pre-processing was conducted using EEGLAB Version 2021.1 (*Delorme and Makeig, 2004*), running in MATLAB Version R2017b. First, EEG data were re-referenced to the mastoid channels. Data were down-sampled to 128 Hz to reduce computation time and memory load required for further pre-processing and analysis. No filtering was applied to data so as not to distort event timing (*van Driel et al., 2019*). Bad channels were noted during data collection and were interpolated using spherical interpolation. On average, 0.49 electrodes were interpolated per recording session. Additionally, one complete session was dropped from further analysis for one participant, due to a poor connection to the mastoid channels. Data were epoched from 100 ms before flash/motion onset to 100 ms after flash/motion offset. The 100 ms period before onset was used to baseline correct each epoch, by subtracting the mean amplitude in this period from the whole epoch.

Eye movement data were used to check fixation: static trials in which gaze deviated more than 2.1 dva from fixation (i.e. was closer to another stimulus position than the central fixation point) at any point while the stimulus was on screen were discarded from analysis, as these eye movements would disrupt retinotopy. On average, 11.2% of trials were rejected on this basis. Participants' eye positions during flashes were further analysed to ensure that there were no systematic eye movements which could be exploited by classifiers during the EEG analysis (see *Figure 2—figure supplement 1*). No motion trials were rejected on the basis of eye movements. This is because motion trials were only used for testing classifiers; if no systematic eye movements are present in the training data, then the classifier cannot learn to distinguish trials on the basis of eye movements, so any eye movements in the test data are irrelevant to the analysis.

Epochs were then automatically rejected through an amplitude threshold. For static trials, epochs were rejected if the standard deviation of the amplitude of any channel exceeded four standard deviations from the mean standard deviation of that channel across all epochs. This resulted in 8.3% of epochs being rejected across all observers. Motion trials were rejected with a threshold of five standard deviations from the mean standard deviation. This less stringent threshold reflects the longer duration of motion trials; more variability in amplitude can be expected; 7.5% of motion trials were rejected across all observers. Finally, static and motion epochs were demeaned. The average amplitude of each electrode across all static stimulus locations was subtracted from each trial amplitude, while for motion trials, the average amplitude from motion vectors of the same length was subtracted. This ensured that the classifiers could leverage any changes in the signal corresponding to stimulus location, without the potential confound of overall amplitude differences in static compared to motion trials (*Dijkstra et al., 2020*).

## EEG analysis

Analyses were programmed using MATLAB Version R2017b and run on the University of Melbourne Spartan High Performance Computing system. Time-resolved multivariate pattern analysis was used to classify EEG data according to the location of the static stimuli. Linear discriminant analysis classifiers with a shrinkage regularisation parameter of 0.05 (*Mostert et al., 2015*) were trained to discriminate the location of static stimuli and tested on unseen static trials or motion trials. Code for classification analysis was adapted from *Mostert et al., 2015*, and *Hogendoorn and Burkitt, 2018*.

To avoid bias that often emerges from multi-class classification (*Yan et al., 2021*), classifiers were trained using pairwise combinations of stimulus positions, such that a classifier was trained to discriminate each location from every other location. As it is redundant to train classifiers to discriminate for example, position 1 vs. 2 and also 2 vs. 1, this resulted in 666 trained classifiers at 90 timepoints over a 350 ms period (from stimulus onset at 0 to 100 ms after stimulus offset). The number of trials in each

class was balanced by sampling trials without replacement from the majority class to equal the number of trials in the minority class.

These classifiers were then tested on every timepoint in either unseen static trials (0–350 ms; fivefold cross-validation between train and test sets) or motion trials (−500 to 500 ms). At each timepoint, pairwise classification results were combined to estimate the likelihood of the stimulus being in a given position (*Manyakov and Van Hulle, 2010*; *Price et al., 1995*). We can estimate $\mathbb{P}(\text{position } i|\text{stimulus is in } s)$ as

$$p_s(i) = \left( \sum_{j \neq i} \frac{1}{r_{ij}} - (k-2) \right)^{-1}. \tag{3}$$

where $r_{ij}$ is the classification performance for position $i$ vs. position $j$, and $k$ is the total number of classes (the 37 stimulus positions, in this case). Such that the probability across all positions was equal to 1, the estimated likelihoods were then normalised between 0 and 1. If decoding performance was at chance level, we would expect uniform likelihood across all stimulus positions, at:

$$\frac{1}{k} \approx 0.027027\ldots \tag{4}$$

A likelihood greater than this indicates a location-specific neural response to the stimulus. An example probability mass function of the likelihood across all stimulus positions can be found in *Figure 3*, step 2. For the main analysis, we investigated the evolution over time of the likelihood of the stimulus being at the presented position, $p_s(s)$, referred to as the *stimulus-position likelihood*. Where relevant, chance level (1/37) was subtracted from the likelihood for easier interpretation in graphs.

In this analysis, we trained classifiers at multiple timepoints because time elapsing post stimulus onset can be seen as a proxy for processing stage. As time passes, stimulus-evoked activity will progress through the visual system (*Dijkstra et al., 2020*; *King and Wyart, 2021*). Our aim was to establish, for each training timepoint, the timepoint in the test data at which the stimulus was most likely to be at a certain position. This tells us the latency of a particular pattern of activity, or representation of the stimulus, in the training data compared to the test data.

We first demonstrated that stimulus position could be discriminated even when static stimuli were close together, by averaging classification results according to distance between stimulus locations (*Figure 2a*). Next, we calculated the latency of representations when training and testing on static stimuli, by taking the maximum stimulus-position likelihood over the test time-window for each training timepoint. This was used as a baseline to which the motion was compared, as the static stimulus locations were unpredictable. Any shifts in latency seen in the motion trials must be due to the predictable preceding trajectory. The key analysis was, therefore, training classifiers to discriminate the location of static stimuli and testing on motion vectors. In this case, in the training data, the stimulus was centred at a certain position, so the timepoint at which the test data is most similar should be the timepoint at which the stimulus was represented in the brain at this position in the trajectory.

For motion trials, to overcome the noise of individual data points, we fit a Gaussian curve with four free parameters to the observed time-course of the calculated likelihood averaged across participants, separately for each training time:

$$p_s(s) = b_1 \exp\left( -\frac{t_{\text{test}} - b_2^2}{2b_3^2} \right) + b_4. \tag{5}$$

The parameter of interest is $b_2$, which describes the horizontal shift of the peak of the Gaussian curve.

## Statistical analysis

Statistical significance of classification results was ascertained through permutation testing. After running the classification analyses as described above, class labels were randomly shuffled when calculating the stimulus-position likelihood, ensuring that the permuted classification results were uninformative about stimulus location. This procedure was repeated 1000 times per participant, providing a null distribution against which our results could be compared with Yuen's $t$-test, one-tailed, $\alpha = 0.05$

(*Yuen, 1974*). Cluster-based correction for multiple comparisons was applied with 1000 permutations (cluster-forming $\alpha$ = 0.05, *Bullmore et al., 1999*; *Maris and Oostenveld, 2007*). Code for the cluster-based correction came from the Decision Decoding Toolbox (*Bode et al., 2019*), which uses code originally from LIMO EEG (*Pernet et al., 2011*) to implement Yuen's *t*-test.

To test significance of linear regression models against a constant model, we used one-tailed *F*-tests. To test whether individual regression coefficients were significantly different from zero, we used two-tailed *t*-test.

## Materials availability

Code is available at https://osf.io/jbw9m/. Processed data is available at https://doi.org/10.5061/dryad.vx0k6djw0 (*Johnson et al., 2022*).

## Acknowledgements

The authors gratefully acknowledge support from the Australian Research Council to HH (DP180102268 and FT200100246). This research was further supported by The University of Melbourne's Research Computing Services and the Petascale Campus Initiative. Thanks to Andrea Titton for discussion on calculating and reporting likelihoods, and to Jane Yook and Vinay Mepani for help with data collection.

## Additional information

### Funding

| Funder | Grant reference number | Author |
| --- | --- | --- |
| Australian Research Council | DP180102268 | Hinze Hogendoorn |
| Australian Research Council | FT200100246 | Hinze Hogendoorn |

The funders had no role in study design, data collection and interpretation, or the decision to submit the work for publication.

### Author contributions

Philippa Anne Johnson, Conceptualization, Data curation, Software, Formal analysis, Validation, Investigation, Visualization, Methodology, Writing – original draft, Project administration, Writing – review and editing; Tessel Blom, Formal analysis, Methodology, Writing – review and editing; Simon van Gaal, Supervision, Writing – review and editing; Daniel Feuerriegel, Methodology, Writing – review and editing; Stefan Bode, Resources, Supervision, Writing – review and editing; Hinze Hogendoorn, Conceptualization, Resources, Formal analysis, Supervision, Funding acquisition, Methodology, Writing – review and editing

### Author ORCIDs

Philippa Anne Johnson http://orcid.org/0000-0002-6125-3138
Simon van Gaal http://orcid.org/0000-0001-6628-4534

### Ethics

Human subjects: Participants gave written informed consent for participation and publication of results before participation. Ethical approval was granted by the University of Melbourne Human Research Ethics Committee (Ethics ID: 1954628.2).

### Decision letter and Author response

Decision letter https://doi.org/10.7554/eLife.82424.sa1
Author response https://doi.org/10.7554/eLife.82424.sa2

## Additional files

### Supplementary files
• MDAR checklist

### Data availability
All data files have been deposited on Dryad at https://doi.org/10.5061/dryad.vx0k6djw0. Code is available on the Open Science Framework at https://osf.io/jbw9m/.

The following datasets were generated:

| Author(s) | Year | Dataset title | Dataset URL | Database and Identifier |
|---|---|---|---|---|
| Johnson PA, Blom T, van Gaal S, Feuerriegel D, Bode S, Hogendoorn H | 2022 | EEG and eyetracking response to static and moving stimuli | https://doi.org/10.5061/dryad.vx0k6djw0 | Dryad Digital Repository, 10.5061/dryad.vx0k6djw0 |
| Johnson P | 2020 | Position representations of moving objects align with real-time position in the early visual response | https://osf.io/jbw9m/ | Open Science Framework, jbw9m |

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
