## [Editor Report]

This article presents compelling visual motion processing data generated via clever analyses, to shed fundamental insights into how the brain compensates for neural processing delays. The approach and findings will be of interest to all neuroscientists – where neural delays prove a sticking point for many theories. One of its major strengths lies in the successful attempt to generalize neural representations of static objects to moving objects.

---

## [Decision Letter]

**Decision letter after peer review:**

Thank you for submitting your article "Position representations of moving objects align with real-time position in the early visual response" for consideration by *eLife*. Your article has been reviewed by 3 peer reviewers, including Clare Press as Reviewing Editor and Reviewer #1, and the evaluation has been overseen by Tirin Moore as the Senior Editor. The following individual involved in the review of your submission has agreed to reveal their identity: Samson Chota (Reviewer #2).

Summary of Revisions:

Overall assessment:

The reviewers find the patterns in the data interesting and believe this view would be shared by other visual neuroscientists as well as neuroscientists from other fields – where neural delays are likely to prove a universal sticking point to many theories. One of its major strengths lies in the successful attempt to generalize neural representations of static objects to moving objects. They also found the manuscript to be well-written – well-tuned, clear, and interesting, with well-presented figures.

Introduction:

1. A little more work could be done to sell this to readers who do not study motion perception in vision. Compensating for neural processing delays will be a problem that most neuroscientific theories need to incorporate; hence why we believe this is especially suitable for publication in *eLife*. We would suggest adding some literature considering an application to other sensory (e..g, auditory) processing, as well as potentially higher-level cognitive domains.

Results:

2. We found the Results section to be well-written, like the rest of the manuscript, but due to the complexity of the analysis we think the reader could be given a little more help. E.g., in the final paragraph of the Results section can some clear flags be inserted to show which aspects of the regression result support which of the conclusions? We think this would help more non-specialised readers grasp the reasons for the conclusion.

3. We would recommend the authors include an illustration of the compensation achieved at different cortical processing stages based on their findings. Perhaps it also makes sense to include approximate neural delays at different stages for a direct comparison. In addition, it might also help to illustrate how this compensation is presumably achieved in space by shifting encoding locations. We believe that these concepts can sometimes be tricky to communicate and the paper would benefit from those visual aids.

4. Have the authors tested for potential differences in how well static positions and movements can be decoded at different locations in the visual field? We can imagine that many readers would be curious if (at least static) decoding is uniform across the visual field. If certain positions/movements are decoded better then the authors could also consider the following analysis: Was there a group of subjects (or selection of best stimulus locations for decoding) for which the early static representations (<100 ms) reliably generalized to motion? This might help to distinguish between poor signal-to-noise and entirely different codes (or direct connections from LGN/pulvinar) between early processing of static and motion stimuli.

Discussion:

5. The findings reported are derived from a specific case of motion perception which may not reflect the general mechanisms optimized for motion perception. The limitations related to task designs and the neural readouts should be discussed as they affect the way that the reported results will be interpreted.

6. This study is clearly different from the authors' earlier work, e.g., Blom et al. However, we think it would be useful to add some discussion to the Discussion section to outline what these Blom et al. studies showed and clearly, therefore, flag the novelty of this study/analysis.

7. The approach of using different epochs to train to approximate different stages of neural processing is clever. However, it seems that later epochs should reflect processing in later hierarchical points while still also reflecting processing in early points in the hierarchy. The authors should discuss the implications of this for their conclusions.

8. Could there be an impact on these findings of a large neural response when a stimulus enters the edge of a neuron's RF, followed by some adaptation? This impact should be discussed.

9. The findings displayed in Figure 4 (reversal of time to peak likelihood) are very intriguing. We believe the manuscript would benefit from discussing these particular results in a bit more detail in the discussion, especially in the light of feedback signals during motion processing. If the observed 'reversed' patterns indeed reflect feedback then by the time the feedback arrives back in the early visual cortex the original stimulus will have moved on and might be replaced by activity from subsequent (mismatching) bottom-up stimulus processing. In addition, referring to Hogendoorn and Burkitt (2019) the authors make a point that delayed compensation should occur in feedback activity, parts of which they do seem to capture with their method (Figure 4). Can the authors elaborate a bit on the potential functional relevance of the observed feedback signals and how they might differ from motion extrapolation-specific feedback?

*Reviewer #1 (Recommendations for the authors):*

1. I think a little more work could be done to sell this to readers who do not study motion perception in vision. Compensating for neural processing delays will be a problem that most neuroscientific theories need to incorporate; hence why I believe this is especially suitable for publication in *eLife*. I would suggest adding some literature considering an application to other sensory (e.g, auditory) processing, as well as potentially higher-level cognitive domains.

2. This study is clearly different from the authors' earlier work, e.g., Blom et al. However, I think it would be useful to add some discussion to the Discussion section to outline what these Blom et al. studies showed and clearly, therefore, flag the novelty of this study/analysis.

3. I found the Results section to be well-written, like the rest of the manuscript, but due to the complexity of analysis, I think the reader could be given a little more help. E.g., in the final paragraph of the Results section can some clear flags be inserted to show which aspects of the regression result support which of the conclusions? I think this would help more non-specialised readers grasp the reasons for the conclusion.

4. Could there be an impact on these findings of a large neural response when a stimulus enters the edge of a neuron's RF, followed by some adaptation?

5. The approach of using different epochs to train to approximate different stages of neural processing is clever. However, it seems that later epochs should reflect processing in later hierarchical points while still also reflecting processing in early points. The authors should discuss the implications of this for their conclusions.

*Reviewer #2 (Recommendations for the authors):*

I honestly have very few remarks on this manuscript all of which concern minor details on form and interpretation. It's fantastic to work, and I am happy to recommend this paper for publication in *eLife*.

Remarks

1. I would recommend the authors include an illustration of the compensation achieved at different cortical processing stages based on their findings. Perhaps it also makes sense to include approximate neural delays at different stages for a direct comparison. In addition, it might also help to illustrate how this compensation is presumably achieved in space by shifting encoding locations. I believe that these concepts can sometimes be tricky to communicate and the paper would benefit from those visual aids.

2. Have the authors tested for potential differences in how well static positions and movements can be decoded at different locations in the visual field? I'm happy for the authors that the analysis works when collapsing across all positions, but I can imagine that many readers would be curious if (at least static) decoding is uniform across the visual field. If certain positions/movements are decoded better then the authors could consider the analysis in point 4 (at their own discretion).

3. The findings displayed in Figure 4 (reversal of time to peak likelihood) are very intriguing. I was hoping the authors could discuss these particular results in a bit more detail in the discussion, especially in the light of feedback signals during motion processing. If the observed 'reversed' patterns indeed reflect feedback then by the time the feedback arrives back in the early visual cortex the original stimulus will have moved on and might be replaced by activity from subsequent (mismatching) bottom-up stimulus processing. This seems to be somewhat problematic for their interpretation. In addition, referring to Hogendoorn and Burkitt (2019) the authors make a point that delay compensation should occur in feedback activity, parts of which they do seem to capture with their method (Figure 4). Can the authors elaborate a bit on the potential functional relevance of the observed feedback signals and how they might differ from motion extrapolation specific feedback?

4. Was there a group of subjects (or selection of best stimulus locations for decoding) for which the early static representations (<100 ms) reliably generalized to motion? This might help to distinguish between poor signal-to-noise and entirely different codes (or direct connections from LGN/pulvinar) between early processing of static and motion stimuli.

*Reviewer #3 (Recommendations for the authors):*

I found the question under investigation to be interesting and I appreciate the authors' choice of recording modality and analytical methods. The paper is also well-organized and the results are presented with clarity. However, I believe the scope of this work is rather narrow and is better suited for a more specialized journal with a focus on human cognition/perception.

---

## [Author Response]

Summary of Revisions (for the authors):Overall assessment:The reviewers find the patterns in the data interesting and believe this view would be shared by other visual neuroscientists as well as neuroscientists from other fields – where neural delays are likely to prove a universal sticking point to many theories. One of its major strengths lies in the successful attempt to generalize neural representations of static objects to moving objects. They also found the manuscript to be well-written – well-tuned, clear, and interesting, with well-presented figures.

We thank the reviewers for their thoughtful, positive and constructive comments on the manuscript. Please find responses to the outstanding questions below. We believe that the edits made while addressing these questions have substantively improved the manuscript.

Introduction:1. A little more work could be done to sell this to readers who do not study motion perception in vision. Compensating for neural processing delays will be a problem that most neuroscientific theories need to incorporate; hence why we believe this is especially suitable for publication in eLife. We would suggest adding some literature considering an application to other sensory (e.g, auditory) processing, as well as potentially higher-level cognitive domains.

Thank you for pointing out the broader application of our work. We certainly agree that the problem of overcoming neural delays is not restricted to motion perception. We have therefore broadened the scope of the introductory paragraphs:

Lines 23-43: “Responding quickly to the world around us is a primary function of the central nervous system: catching prey, escaping predators, and avoiding moving objects (e.g., falling rocks) are all crucial to survival. […] In this case, there is evidence that predictive processes help to compensate both for the neural delays incurred before visual input reaches the visual cortex and for the delays incurred during subsequent cortical processing (Berry et al., 1999; Jancke et al., 2004; Orban et al., 1985; Subramaniyan et al., 2018; Sundberg et al., 2006).”

Results:2. We found the Results section to be well-written, like the rest of the manuscript, but due to the complexity of the analysis we think the reader could be given a little more help. E.g., in the final paragraph of the Results section can some clear flags be inserted to show which aspects of the regression result support which of the conclusions? We think this would help more non-specialised readers grasp the reasons for the conclusion.

Thank you for this comment. We have added some pointers to the specific equations that support our conclusions in the final section of the Results (see below), as well as rephrasing some sections throughout the Results, which we hope will improve clarity. We have also added an additional statistical test that enables us to directly compare the gradient and intercept of the relationship between training time and peak times for static and motion stimuli, which we hope makes it clearer from where our conclusions are coming. Additionally, please see the response to Question 3 for a new figure, which we hope helps illustrate our findings.

Results, Lines 260-290: “Furthermore, the difference between the static and motion gradients was not significantly different from zero (*t* = -1.58, *p* = 0.13) but the difference between the two intercepts was (*t* = 5.32, *p* = 3.32x10^-5^).[…] Based on the training time, these early stimulus representations likely originated in early visual cortex (V1-3), meaning that the early visual system was able to almost completely compensate for neural delays accumulated during processing up to that point and represent moving objects close to their real-time position.”

3. We would recommend the authors include an illustration of the compensation achieved at different cortical processing stages based on their findings. Perhaps it also makes sense to include approximate neural delays at different stages for a direct comparison. In addition, it might also help to illustrate how this compensation is presumably achieved in space by shifting encoding locations. We believe that these concepts can sometimes be tricky to communicate and the paper would benefit from those visual aids.

Thank you for this suggestion. An additional panel has been added to Figure 4, as well as text to explain it in the final section of the Results. We hope this illustration helps to communicate the delays that are compensated at different levels of processing. We chose not to illustrate the spatial lag, but have added additional text to explain how the temporal shift corresponds to a spatial shift.

Lines 266-268: In other words, delays that accumulate during cortical processing did not appear to be compensated when processing motion. This is illustrated in Figure 4c, where the relative delay between neural representations is preserved regardless of whether the stimulus is moving or not.

Lines 280-287: In turn, this means that these neural representations were activated at the time that the moving object was physically centred on the corresponding position. This can also be seen in Figure 4c: the representations of the moving stimulus emerge 73ms earlier than the same representations of the flashed stimulus. The earliest representation of the moving stimulus peaks when t = 0ms, the time at which the moving stimulus is in the same position of the flash. This corresponds to a spatial shift in the encoding of the moving stimulus, as, by the time the representations of the flash emerge, the moving stimulus will be represented further ahead on its path.

4. Have the authors tested for potential differences in how well static positions and movements can be decoded at different locations in the visual field? We can imagine that many readers would be curious if (at least static) decoding is uniform across the visual field. If certain positions/movements are decoded better then the authors could also consider the following analysis: Was there a group of subjects (or selection of best stimulus locations for decoding) for which the early static representations (<100 ms) reliably generalized to motion? This might help to distinguish between poor signal-to-noise and entirely different codes (or direct connections from LGN/pulvinar) between early processing of static and motion stimuli.

We had not tested this, but we agree that these aspects of the decoding analysis are worth exploring further.

Firstly, we have investigated how stimulus-position likelihood varies over the visual field for the static stimulus presentations. We have now included the graph as a supplement to Figure 2. It shows the difference from chance likelihood as a function of time, with different lines corresponding to stimuli at different eccentricities. We decided to average by eccentricity, as the graph is quite difficult to read with 37 lines (i.e. one for each stimulus position; this can be seen in Figure II, top left panel).

It can be observed that decoding performance is highest for stimuli at fixation, and that performance generally decreases with increasing eccentricity. However, such differences are not clearly visible for the three furthest eccentricities relative to fixation. We believe that this is because, although the most eccentric stimuli probably elicit the smallest EEG signal, these likelihood calculations include comparisons between stimuli that are very far apart. For example, some pairwise comparisons included in the likelihood calculation for the stimulus on the far left are up to 24dva apart, but at fixation the maximum distance between the stimulus and the comparison positions is half the grid, 12dva. In Figure 2A in the manuscript, it can be seen that classification performance is highest for pairwise comparisons between stimuli further apart. Additionally, it has been shown that foveally presented stimuli elicit a much larger event-related potential (ERP) than peripheral stimuli (Rousselet et al., 2005), which would likely result in larger-magnitude ERPs for static stimuli at fixation. This could explain why the likelihood is dramatically higher for stimuli presented at fixation. We have noted these issues in the Figure 2 supplement caption.

Results, Lines 112-114: Stimulus-position likelihoods by eccentricity (distance from fixation) are plotted in Figure 2 —figure supplement 1.

Secondly, we explored whether selecting certain participants/positions with better position decoding would enable us to investigate a wider time window of motion processing. To this end, we selected good participants or good positions on the basis of the strength of decoding of flashes (to avoid cherry-picking within the motion decoding). More specifically, we selected either participants (Author response image 1) or positions (Author response image 2) for which decoding of the position of the flash was above 2*chance likelihood (i.e. 2*1/37). Through this approach we selected three participants and six positions. With regards to position, the best decoding of stimulus location was possible when the stimulus was horizontally aligned with, or just below, fixation. This is consistent with known anisotropies in visual field processing and representation (e.g. Carrasco et al., 2001; Van Essen et al., 1984). This selection did not enable us to meaningfully extend our results to earlier timepoints and, due to the exclusion of significant amounts of data, we believe our signal-to-noise ratio was lowered. In the case of looking at the best positions, we can now fit a Gaussian with R^2^ greater than 0.5 at training timepoints before 50ms training time, when it is impossible that stimulus information is present in the brain. This indicates that the selection of a subset of participants and positions may be biasing our results toward finding decodable information at times when this would not be available to participants in the flashed condition.

One point of interest is that with both of these selection methods, the time of peak likelihood for training times 80-100ms is well before 0ms. This implies that these early representations of the stimulus might be over-extrapolated and that the brain may be representing the stimulus in a position it has not yet reached. However, we are reluctant to overinterpret analyses that lack power and as such we would prefer not to include these additional analyses in the main manuscript.

**Author response image 1. sa2fig1:** Selection of participants for whom flash decoding is best. Participants were selected if the difference from chance stimulus-position likelihood (averaged across all stimulus positions) was above a threshold of 1/37 (which is the same as the likelihood being twice as high as chance level). The top left panel shows the difference from chance likelihood against training time. The three selected participants are shown in a darker colour. The remaining panels show graphs corresponding to the graphs in the main manuscript: the analysis is exactly the same.

**Author response image 2. sa2fig2:** Selection of positions for which flash decoding is best. Positions were selected if the difference from chance stimulus-position likelihood (averaged across all participants) was above a threshold of 1/37. In the top left panel, lines show difference from chance likelihood for all of the 37 possible stimulus locations. Inset on the top left panel shows the selected positions, highlighted in red (fixation) and black on the graph. Everything else is the same as above.

Discussion:5. The findings reported are derived from a specific case of motion perception which may not reflect the general mechanisms optimized for motion perception. The limitations related to task designs and the neural readouts should be discussed as they affect the way that the reported results will be interpreted.

We agree that these findings reflect just one specific case of motion, in particular because the stimulus moved at a constant (rather slow) speed, and along a predictable path. We have added the following sections to the Discussion to ensure this limitation is explicit. Some limitations of the neural readout approach are already discussed, namely the inability to pinpoint the signal to a particular neural source, and the low signal-to-noise ratio. To flag this section more clearly, and to better organise the Discussion in general, we have added subheadings to the Discussion. This section is now titled ‘Limitations’. We would welcome any further suggestions of limitations that we should include in this section. We have also compared the smooth motion used in our study to apparent motion, and have highlighted important differences between these, in our response to comment 6.

Lines 364 to 369: “If information about object motion is used to compensate for delays via feedback connections, it could be expected that the particular speed of the stimulus would be exploited to maximise the accuracy of the position representation. In the present study, we only tested one stimulus speed, so further research including a range of speeds, as found in natural scenes (Wojtach et al., 2008), is necessary to examine whether compensatory mechanisms work in this way.”

Lines 415 to 417: “Future research should extend the present findings by investigating the neural consequences of an unexpected change of direction, and generally unpredictable motion, in a smooth motion trajectory.”

6. This study is clearly different from the authors' earlier work, e.g., Blom et al. However, we think it would be useful to add some discussion to the Discussion section to outline what these Blom et al. studies showed and clearly, therefore, flag the novelty of this study/analysis.

Thank you for pointing this out. We have added a paragraph to the discussion comparing our results with Hogendorrn and Burkitt's (2018) and Blom et al.’s (2020; 2021) previous findings regarding apparent motion. Importantly, this prior work differs from ours in that we investigated smooth (rather than apparent) motion in the manuscript. The primary difference between the previous findings and ours is the extent of the compensation found. We have outlined these discrepancies and the possible reasons for differences in findings below:

Lines 304 – 420: “Comparison between smooth and apparent motion

Previous EEG research has investigated whether neural delays are compensated when viewing apparent motion. […] Delays seem to be compensated to a greater extent in this case of smooth motion, so the visual system would have to employ a larger correction for erroneous position encoding if the stimulus changed direction.”

7. The approach of using different epochs to train to approximate different stages of neural processing is clever. However, it seems that later epochs should reflect processing in later hierarchical points while still also reflecting processing in early points in the hierarchy. The authors should discuss the implications of this for their conclusions.

We agree that later timepoints in the epoch could capture ongoing processing at early points in the visual hierarchy, as well as processing in later hierarchical stages. Previous research suggests that the early visual response is dominated by a feedforward cascade, due to the timing of neural responses and delays between activity in different cortical areas (Lamme and Roelfsema, 2000). Additionally, the TGM in Figure 2c shows that there is not substantial cross-generalisation across time before 150ms (i.e. the red blob is quite restricted to the diagonal). Because our regression analyses were restricted to multivariate patterns occurring less than 150 post stimulus onset, we therefore believe this is not a significant issue in the present research. We have now added a sentence to the Results about this issue. However, even if there is lingering activity from earlier visual areas, we would argue that this does not alter the main conclusions of the paper (processing happens at the same rate regardless of whether the stimulus is moving or not; there is a lot of compensation for delays very early in processing). This is now mentioned in the Limitations section of the Discussion.

Results, Lines 130-132: “It appears that the stimulus representations before 150ms training/test time did not generalise to other timepoints, but later reactivation of representations is apparent after 150ms (off-diagonal red blobs).”

Discussion, Lines 435-440: “While the initial response is likely to be dominated by a feedforward cascade through the visual hierarchy (Lamme and Roelfsema, 2000), it is possible that activity recorded at later timepoints in the training epoch reflects ongoing processing in hierarchically early visual areas, as well as additional processing in hierarchically later visual areas. However, this issue does not alter our conclusions concerning the relative timing of activity when viewing static or moving stimuli.”

Relatedly, we agree with the suggestion that later timepoints within the training epoch could reflect reactivation of earlier stimulus representations, as well as further processing of the stimulus. We believe this is the reason that the observed oscillatory pattern emerges. For more discussion of this point, please refer to our response to Question 9.

8. Could there be an impact on these findings of a large neural response when a stimulus enters the edge of a neuron's RF, followed by some adaptation? This impact should be discussed.

Thank you for this suggestion. We agree that adaptation in response to the leading edge of the stimulus is a possible mechanism for the observed effect. This idea is similar to theories about how delays are compensated in the retina when viewing motion (e.g., Berry et al., 1999). Previous studies have suggested that similar computation could indeed occur in the visual cortex, as the organisation of neurons could bring about the balance of excitation and inhibition necessary to compensate for delays in this way (Johnston and Lagnado, 2015). It takes the stimulus 400ms to move from one flash position to the next, so the leading edge of the stimulus is closer to a particular flash position than the previous at -200ms. We see that the likelihood of the stimulus being present in that position starts ramping up much earlier that -200ms, so we think that this compensation is unlikely to be completely driven by a large response to the leading edge, followed by adaptation. We have added words to this effect on Lines 267-273, as shown below:

Lines 347 – 356: “In particular, the retinal compensation mechanisms only appear to act up to speeds of about 5dva/s (Berry et al., 1999; Jancke et al., 2004). This is thought to be achieved by a large response to the leading edge of the stimulus, followed by gain control mechanisms reducing neuronal firing rates. In the present study, it takes 400ms for the stimulus to travel from one static stimulus position to the next, so the leading edge of the moving stimulus is closest to the stimulus position it is approaching 200ms before it reaches that position. In general, we see a ramping in the likelihood earlier than -200ms (see Figure 3 – Supplementary Figure 1), so we believe that this adaptation is unlikely to be the sole mechanism responsible for the observed shift in encoded location.”

9. The findings displayed in Figure 4 (reversal of time to peak likelihood) are very intriguing. We believe the manuscript would benefit from discussing these particular results in a bit more detail in the discussion, especially in the light of feedback signals during motion processing. If the observed 'reversed' patterns indeed reflect feedback then by the time the feedback arrives back in the early visual cortex the original stimulus will have moved on and might be replaced by activity from subsequent (mismatching) bottom-up stimulus processing. In addition, referring to Hogendoorn and Burkitt (2019) the authors make a point that delayed compensation should occur in feedback activity, parts of which they do seem to capture with their method (Figure 4). Can the authors elaborate a bit on the potential functional relevance of the observed feedback signals and how they might differ from motion extrapolation-specific feedback?

Thank you for this comment; we agree that feedback activity likely plays an important role in compensation for delays. However, while we do find evidence of continuing feedforward and feedback sweeps of activity, we do not think that this is particularly informative to the research question. This is because, rather than corresponding to further processing of the moving stimulus, we believe the oscillatory pattern emerges because part of the flash-evoked signal reflects reactivation of earlier stimulus representations. To explain further, as time passes after the onset of a static stimulus, stimulus-evoked activity progresses through different visual areas. However, there is also an increase in recurrent processing, whereby earlier representations of the stimulus are reactivated. We believe that after ~150ms, the further processing of the static and moving stimuli diverges. However, because part of the flash-evoked signal reflects recurrent processing in earlier visual areas, it is still possible to decode the position of the motion on the basis of these reactivated representations. We therefore think this oscillatory pattern (which is similar to that seen in Dijkstra et al. (2020)) is more informative about processing of static stimuli than motion. We have added a few sentences to this effect in the Discussion section about feedback. We additionally realised that this line of reasoning was not clear in the Results, so we have tried to clarify this point there as well.

Results, Lines 199-219: **“**This pattern subsequently reverses between 150 and 200ms, indicating that hierarchically later representations were activated at a shorter latency. Finally, from a training time of ~250ms the time to peak likelihood was stable at approximately 50ms. […] If this later activity (>150ms training time) does indeed reflect feedback processing of the static stimulus, then, for these later timepoints, the latency measure we have calculated might not be informative about the time necessary to first represent the moving object, because the initial activations and the reactivations are indistinguishable.”

Discussion, Lines 383 – 391**: “**Although this analysis captures feedforward and feedback sweeps of activity through the visual cortex, we believe that these oscillations are present in the processing of the flash, not the motion. This would mean that the portion of the later flash-evoked signal that cross-generalises to motion reflects reactivation of earlier stimulus representations rather than further stimulus processing. However, there remains the possibility that these oscillations are evoked by motion. This would imply that, during feedback activity, the position of the stimulus is not updated as the stimulus moves, and that the stimulus is likely represented in multiple locations concurrently in the visual system (King and Wyart, 2021). Further research is needed to tease apart these options.”

References

Benvenuti, G., Chemla, S., Boonman, A., Perrinet, L., Masson, G. S., and Chavane, F. (2020). Anticipatory responses along motion trajectories in awake monkey area V1. *BioRXiv*. https://doi.org/10.1101/2020.03.26.010017

Berry, M. J., Brivanlou, I. H., Jordan, T. A., and Meister, M. (1999). Anticipation of moving stimuli by the retina. *Nature*, *398*(6725), 334–338. https://doi.org/10.1038/18678

Blom, T., Bode, S., and Hogendoorn, H. (2021). The time-course of prediction formation and revision in human visual motion processing. *Cortex*, *138*, 191–202. https://doi.org/10.1016/j.cortex.2021.02.008

Blom, T., Feuerriegel, D., Johnson, P., Bode, S., and Hogendoorn, H. (2020). Predictions drive neural representations of visual events ahead of incoming sensory information. Proceedings of the National Academy of Sciences of the United States of America, 117(13). https://doi.org/10.1073/pnas.1917777117

Carrasco, M., P.Talgar, C., and Cameron, E. L. (2001). Characterizing visual performance fields: Effects of transient covert attention, spatial frequency, eccentricity, task and set size. *Spatial Vision*, *15*(1), 61–75. https://doi.org/10.1163/15685680152692015

Chemla, S., Reynaud, X. A., Di Volo, M., Yann Zerlaut, X., Perrinet, X. L., Destexhe, A., and Chavane, X. F. (2019). Suppressive traveling waves shape representations of illusory motion in primary visual cortex of awake primate. *Journal of Neuroscience*, *39*(22). https://doi.org/10.1523/JNEUROSCI.2792-18.2019

Clark, A. (2013). Whatever next? Predictive brains, situated agents, and the future of cognitive science. *Behavioral and Brain Sciences*, *36*(3), 181–204. https://doi.org/10.1017/S0140525X12000477

Dijkstra, N., Ambrogioni, L., Vidaurre, D., and van Gerven, M. (2020). Neural dynamics of perceptual inference and its reversal during imagery. *ELife*, *9*. https://doi.org/10.7554/*eLife*.53588

Friston, K. (2010). The free-energy principle: A unified brain theory? *Nature Reviews Neuroscience*, *11*(2), 127–138. https://doi.org/10.1038/nrn2787

Friston, K. J., Daunizeau, J., Kilner, J., and Kiebel, S. J. (2010). Action and behavior: A free-energy formulation. *Biological Cybernetics*, *102*(3), 227–260. https://doi.org/10.1007/s00422-010-0364-z

Guo, K., Robertson, R. G., Pulgarin, M., Nevado, A., Panzeri, S., Thiele, A., and Young, M. P. (2007). Spatio-temporal prediction and inference by V1 neurons. *European Journal of Neuroscience*, *26*(4), 1045–1054. https://doi.org/10.1111/j.1460-9568.2007.05712.x

Harvey, B. M., and Dumoulin, S. O. (2011). The relationship between cortical magnification factor and population receptive field size in human visual cortex: Constancies in cortical architecture. *Journal of Neuroscience*, *31*(38), 13604–13612. https://doi.org/10.1523/JNEUROSCI.2572-11.2011

Hogendoorn, H., and Burkitt, A. N. (2018). Predictive coding of visual object position ahead of moving objects revealed by time-resolved EEG decoding. *NeuroImage*, *171*, 55–61. https://doi.org/10.1016/j.neuroimage.2017.12.063

Jancke, D., Erlhagen, W., Schöner, G., and Dinse, H. R. (2004). Shorter latencies for motion trajectories than for flashes in population responses of cat primary visual cortex. *J Physiol*, *556*, 971–982. https://doi.org/10.1113/jphysiol.2003.058941

Johnston, J., and Lagnado, L. (2015). General features of the retinal connectome determine the computation of motion anticipation. *ELife*, *2015*(4). https://doi.org/10.7554/*eLife*.06250

Kiebel, S. J., Daunizeau, J., and Friston, K. J. (2008). A Hierarchy of Time-Scales and the Brain. *PLoS Computational Biology*, *4*(11), e1000209. https://doi.org/10.1371/journal.pcbi.1000209

King, J.-R., and Wyart, V. (2021). The Human Brain Encodes a Chronicle of Visual Events at each Instant of Time thanks to the Multiplexing of Traveling Waves. *The Journal of Neuroscience*. https://doi.org/10.1523/JNEUROSCI.2098-20.2021

Lamme, V. A. F. F., and Roelfsema, P. R. (2000). The distinct modes of vision offered by feedforward and recurrent processing. *Trends in Neurosciences*, *23*(11), 571–579. https://doi.org/10.1016/S0166-2236(00)01657-X

Millidge, B., Seth, A., and Buckley, C. L. (2022). Predictive Coding: A Theoretical and Experimental Review. *ArXiv*. https://doi.org/10.48550/arXiv.2107.12979

Nijhawan, R. (1994). Motion extrapolation in catching. *Nature*, *370*(6487), 256–257. https://doi.org/10.1038/370256b0

Orban, G. A., Hoffmann, K. P., and Duysens, J. (1985). Velocity selectivity in the cat visual system. I. Responses of LGN cells to moving bar stimuli: A comparison with cortical areas 17 and 18. *Journal of Neurophysiology*, *54*(4), 1026–1049. https://doi.org/10.1152/jn.1985.54.4.1026

Rousselet, G. A., Husk, J. S., Bennett, P. J., and Sekuler, A. B. (2005). Spatial scaling factors explain eccentricity effects on face ERPs. *Journal of Vision*, *5*(10), 755–763. https://doi.org/10.1167/5.10.1

Subramaniyan, M., Ecker, A. S., Patel, S. S., Cotton, R. J., Bethge, M., Pitkow, X., Berens, P., and Tolias, A. S. (2018). Faster processing of moving compared with flashed bars in awake macaque V1 provides a neural correlate of the flash lag illusion. *Journal of Neurophysiology*, *120*(5), 2430–2452. https://doi.org/10.1152/jn.00792.2017

Sundberg, K. A., Fallah, M., and Reynolds, J. H. (2006). A motion-dependent distortion of retinotopy in area V4. *Neuron*, *49*(3), 447–457. https://doi.org/10.1016/j.neuron.2005.12.023

Van Essen, D. C., Newsome, W. T., and Maunsell, J. H. R. (1984). The visual field representation in striate cortex of the macaque monkey: Asymmetries, anisotropies, and individual variability. *Vision Research*, *24*(5), 429–448. https://doi.org/10.1016/0042-6989(84)90041-5

Wojtach, W. T., Sung, K., Truong, S., and Purves, D. (2008). An empirical explanation of the flash-lag effect. Proceedings of the National Academy of Sciences of the United States of America, 105(42), 16338–16343. https://doi.org/10.1073/pnas.0808916105